# The Influence of Bisphenol a on the Nitrergic Nervous Structures in the Domestic Porcine Uterus

**DOI:** 10.3390/ijms21124543

**Published:** 2020-06-26

**Authors:** Liliana Rytel, Slawomir Gonkowski

**Affiliations:** 1Department of Internal Disease with Clinic, Faculty of Veterinary Medicine, University of Warmia and Mazury, Street Oczapowskiego 14, 10-719 Olsztyn, Poland; 2Department of Clinical Physiology, Faculty of Veterinary Medicine, University of Warmia and Mazury, Street Oczapowskiego 14, 10-719 Olsztyn, Poland; slawekg@uwm.edu.pl

**Keywords:** bisphenol A, nitric oxide, innervation, neuropeptides, uterus

## Abstract

Bisphenol A (BPA) is one of the most common environmental pollutants among endocrine disruptors. Due to its similarity to estrogen, BPA may affect estrogen receptors and show adverse effects on many internal organs. The reproductive system is particularly vulnerable to the impact of BPA, but knowledge about BPA-induced changes in the innervation of the uterus is relatively scarce. Therefore, this study aimed to investigate the influence of various doses of BPA on nitrergic nerves supplying the uterus with the double immunofluorescence method. It has been shown that even low doses of BPA caused an increase in the number of nitrergic nerves in the uterine wall and changed their neurochemical characterization. During the present study, changes in the number of nitrergic nerves simultaneously immunoreactive to substance P, vasoactive intestinal polypeptide, pituitary adenylate cyclase-activating peptide, and/or cocaine- and amphetamine-regulated transcript were found under the influence of BPA. The obtained results strongly suggest that nitrergic nerves in the uterine wall participate in adaptive and/or protective processes aimed at homeostasis maintenance in the uterine activity under the impact of BPA.

## 1. Introduction

Nitric oxide (NO) is one of the gaseous neurotransmitters and/or neuromodulators, which is widely distributed in living organisms [1]. NO is synthesized from L-arginine in reactions catalyzed by nitric oxide synthase, existing in three isoforms: brain or neuronal (bNOS or nNOS), endothelial (eNOS), and inducible calcium-independent (iNOS) [2]. Among these isoforms, nNOS is widely regarded as a marker of neurons synthesizing NO [3]. To date, the presence of NO has been reported in numerous nervous structures in both the central and peripheral nervous systems [4]. In the light of the previous studies, it is known that NO may play a variety of functions within the nervous system both in physiological conditions and under the impact of pathological factors and largely depends on the part of the nervous system and the organs supplied by it [5]. 

One of the organs in which NO plays an important role is the uterus. Previous studies have reported that NO in the uterus may be located in the nerve fibers, uterine muscular cells, endometrium, and the wall of vessels [6]. It is also known that NO in the uterus may show multidirectional activity. It is primarily involved in the regulation of the activity of the uterine muscular layer, causing the relaxation of the smooth muscles, which is especially visible during pregnancy [7]. It is also known that NO in the uterus acts as a strong, important vasodilator and affects endometrial secretion, which can be important at implantation time [8]. 

As regards the nitrergic nerve fibers located in the uterine wall, it is known that such fibers are relatively numerous and distributed in various parts of the organ, both in the muscular layer and endometrium [9]. Previous studies have also reported that nitrergic uterine nerves show neurochemical diversity and may also contain various other neuronal active substances, including tyrosine hydroxylase, vesicular acetylcholine transporter substance P, vasoactive intestinal polypeptide, and/or calcitonin gene-related peptide [10,11]. The wide range of neuronal factors that are present in the nitrergic nerves indicates that uterine fibers containing NO play multidirectional roles in the regulation of the uterus activity, not only involving muscular functionality and endometrial secretion but also processes connected with intrauterine blood flow and sensory stimuli conduction [12,13]. Moreover, it is known that nitrergic uterine fibers take part in pathological states involving this organ, which is most likely associated with the important roles of nerves containing NO in oxidative stress modulation [14].

However, it should be noted that several aspects connected with distribution, neurochemical characterization of uterine nitrergic nerves, as well as their participation in mechanisms connected with the impact of toxic factors remain not fully elucidated. In particular, little is known about the distribution and functions of uterine nitrergic nerves in the domestic pig which, due to similarity of its autonomic nervous system to humans in terms of biochemical and neurochemical organization, is considered to be an optimal animal model for studies reflecting the influence of pathological and toxic factors on the human nervous system [15].

Therefore, the present study aimed to investigate the influence of bisphenol A (BPA) on the uterine nitrergic neurons located in the various parts of the wall of the porcine uterus. BPA is an organic substance used for decades in the production of plastics and epoxy resins. It is present in many objects of everyday use, such as toys, furnishing, elements of cars, and even dental materials [16]. However, objects containing BPA coming into contact with food or drinking water (including food containers and bottles) are the greatest threat, because this substance may be released from plastics, penetrate food, and enter living organisms [17]. Due to its similarity to estrogen, BPA may affect the estrogen receptors widely distributed in the body and cause dysfunctions of various internal organs and systems [18]. For this reason, BPA is listed as a potent endocrine disruptor [19].

One of the internal organs which, due to the occurrence of a high number of estrogen receptors [20], is especially vulnerable to adverse effects of BPA is the uterus. Previous studies have reported that exposure to BPA leads to changes within the endometrium, which are manifested by fluctuations in the number of endometrial fibroblasts, inhibition of apoptotic reactions, and an increase in the thickness of this part of the uterine wall [10,11]. Moreover, BPA causes inhibition of the activity of the uterine muscular layer [21], distorts the estrus cycle, affects pregnancy [22], and disrupts embryo implantation in the uterus [23]. Some studies have reported the correlations between excessive exposure to BPA and diseases developing in the reproductive organs, including stromal polyps, endometriosis, and cervical cancer [24].

One of the most interesting aspects of the impact of BPA on the uterus is connected with the influence of this substance on the nervous system. Although it is known that neuronal tissue is sensitive to BPA and changes caused by this substance have been observed in various parts in both the central and peripheral nervous systems [25,26], these changes consist of disturbances in the development of dendrites and axons, changes in synaptogenesis, abnormalities in intraneuronal calcium ion transport, changes in neurochemical characterization of neuronal cells and nerve fibers, and disturbances in higher neuronal functions, including memory and learning [27]. Moreover, some investigations connect a high degree of exposure to BPA with neurodegenerative disorders, such as Alzheimer’s or Parkinson’s diseases [28,29]. Although it is known that even low doses of BPA may result in changes in the expression of neuronal active substances within the nerve fibers located in the uterine wall [10], it should be underlined that knowledge concerning this issue is extremely fragmentary. 

Therefore, the present study is not only a comprehensive description of the distribution and neurochemical characterization of uterine nitrergic nerve fibers in the domestic pig in physiological conditions and under the impact of BPA but will also be the first step to a better understanding of the mechanisms connected with the influence of BPA on the human reproductive system. 

## 2. Results

This study indicated the presence of nNOS-like immunoreactive (nNOS-LI) nerve fibers in the corpus and horns of the porcine uterus in both control animals and pigs treated with both doses of BPA, and such fibers were located in the endometrium and myometrium (Table 1 and Table 2). Moreover, nerves immunoreactive to nNOS also showed the presence of pituitary adenylate cyclase-activating peptide (PACAP), vasoactive intestinal polypeptide (VIP), substance P (SP), and/or cocaine- and amphetamine-regulated transcript (CART) (Figure 1, Figure 2, Figure 3 and Figure 4).

### 2.1. Nitrergic Nerves in the Porcine Uterus under Physiological Conditions

#### 2.1.1. The Uterine Corpus

In the uterine corpus of control animals, the number of nNOS-LI nerves in the endometrium averaged 5.19 nerve fibers per microscopic observation field. In the myometrium, this value was similar and reached 4.74. The largest percentage of nitrergic nerves located in the endometrium of the uterine corpus simultaneously contained PACAP and/or SP (Table 2 and Table 3). These substances were noted in 32.56% and 31.94% of all nerve fibers immunoreactive to nNOS, respectively. The degree of co-localization of nNOS with other substances was slightly lower. The percentage of nNOS+/VIP+ nerves amounted to 29.07%, and nNOS+/CART+ reached 29.2% of all nNOS-LI nerve fibers. In the myometrium of the uterine corpus, the largest number of nitrergic nerves were also immunoreactive to SP, which was found in 40.33% of all nNOS-LI nerves. A slightly lower percentage of nitrergic nerves contained VIP (34.91%) and/or PACAP (30.83%), and the lowest number of nNOS-LI fibers showed the presence of CART, which was noted in 24.67% of all nitrergic nerves. 

#### 2.1.2. The Uterine Horns

During the present study, the number of nitrergic nerves and their neurochemical characterization in both uterine horns were similar. The average number of nNOS-LI nerves in the uterine horns was much lower than that noted in the corpus. In the endometrium, these values amounted to 1.54 fibers per observation field in the right horn and 1.65 in the left horn (Table 2). In turn, the number of nitrergic nerves in the myometrium was 1.54 and 1.66 in the right and left horn, respectively. The largest number of nitrergic nerves located in the endometrium of the uterine horns also showed the presence of SP and/or PACAP. SP was found in 31.05% of nitrergic nerves located in the endometrium of the right horn and 31.56% of such nerves in the left horn. In the case of PACAP, these values amounted to 26.42% and 25.52%, respectively. A slightly lower number of nNOS-LI nerves also showed immunoreactivity to VIP (24.33% in the right horn and 25.21% in the left horn), and the percentage of nNOS+/CART+ nerves was the smallest (19.04% and 18.74%, respectively). The neurochemical characterization of nitrergic nerves located in the myometrium of the uterine horns was similar to that noted in the endometrium. In particular, the largest number of nNOS-LI nerves was also immunoreactive to SP (30.44% of all nNOS-LI nerves in the right horn and 30.53% in the left horn). The presence of PACAP and/or VIP was noted in the slightly lower number of nitrergic nerves. In the case of PACAP, these values amounted to 28.5% in the right horn and 27.22% in the left horn, and in the case of VIP, it was 25.97% and 25.99%, respectively. The lowest number of nitrergic nerves in the myometrium of the uterine horns simultaneously showed the presence of CART (26.11% in the right horn and 25.43% in the left horn).

### 2.2. The Influence of BPA on Nitrergic Nerves in the Porcine Uterus

During the present study, the impact of BPA in both used doses on the number and neurochemical characterization of nitrergic nerves located in the porcine uterus was observed. Generally (with a few exceptions), BPA caused an increase in the number of nitrergic nerves and an increase in the degree of co-localization of nNOS with other substances studied. Moreover, the observed changes were more visible under the impact of higher doses of BPA (Figure 1, Figure 2, Figure 3 and Figure 4).

#### 2.2.1. The Uterine Corpus

In the endometrium of the uterine corpus, the number of nitrergic nerves in animals treated with low doses of BPA amounted to 4.71 fibers per observation field and this value showed no statistically significant differences in comparison with the control pigs (Table 3). In turn, in animals that received high doses of BPA, the number of nitrergic nerves was two and a half times higher than that noted in the control animals and averaged 13.07 nerves per observation field. The situation was similar in the myometrium, where the number of nNOS-LI nerves under the impact of low doses of BPA reached 5.52, and in animals that received high doses of BPA, it reached 13.5 nerves per observation field. 

Both doses of BPA changed the neurochemical characterization of nNOS-LI nerves located in the endometrium and myometrium of the uterine corpus (Table 1 and Table 2). In the endometrium, the most visible increase was noted in the percentage of nitrergic nerves, which simultaneously contained VIP and/or PACAP. The percentage of nNOS+/VIP+ nerves in animals receiving low doses of BPA reached 44.46% of all nNOS-LI fibers, and in animals treated with high doses of BPA had more than twice as much as that noted in the control animals and it amounted to 64.13%. In turn, PACAP was found in 42.11% of all nNOS-LI nerves in animals treated with low doses of BPA and in 60.27% of such nerves receiving high doses of this substance. The changes concerning co-localization of nNOS with SP were less visible. SP was present in 39.1% of nNOS-LI nerves in animals treated with low doses of BPA and in 50.64% of such nerves in pigs exposed to high doses of this substance. The least visible changes concerned nNOS+/CART+ nerves. Their number increased to 31.92% of all nNOS-LI nerves under the impact of low doses of BPA and to 36.08% under the impact of high doses of this substance.

BPA-induced changes in the neurochemical characterization of nitrergic nerves located in the myometrium of the uterine corpus were similar to those noted in the endometrium (Table 2 and Table 3). The most visible increase concerned nNOS+/VIP+ and nNOS+/PACAP+ nerves. The first of them amounted to 46.22% of all nNOS-LI nerves in animals treated with low doses of BPA and 62.53% in pigs under the impact of high doses of BPA. In turn, PACAP was found in 48.65% and 57.78% of nNOS-LI nerves in animals treated with low and high doses of BPA, respectively. The influence of BPA on the degree of co-localization of nNOS with SP and/or CART was smaller. The low and high doses of BPA caused an increase in the number of nNOS+/SP+ to 44.76% and 54.74%, respectively. In turn, the percentage of nitrergic nerves simultaneously containing CART reached 32.46% in animals treated with low doses of BPA and 36.25% in pigs under the impact of high doses of this substance.

#### 2.2.2. The Uterine Horns

The character and severity of BPA-induced changes in the number and neurochemical characterization of nitrergic nerves were similar in the right and left uterine horn. Both low and high doses of BPA caused an increase in the number of nitrergic nerves located in the endometrium and myometrium (Table 1 and Table 2) of the uterine horns. 

Under the impact of low doses of BPA, the number of nNOS-LI nerves in the endometrium increased to 3.43 nerves per observation field in the right horn and to 3.63 in the left horn. High doses of BPA caused even more visible changes. The number of nNOS-LI nerves in animals treated with high doses of BPA was about three times higher than the number noted in control animals and was 4.17 nerves per observation field in the right horn and 4.38 in the left horn.

The administration of both doses of BPA resulted in an increase in the degree of co-localization of nNOS with all substances investigated within nerves located in the endometrium of the uterine horns (Table 1). The most visible changes concerned nNOS+/VIP+ nerves, whose number increased under the impact of low doses of BPA to 35.28% of all nNOS-LI nerves in the right horn and to 36.51% in the left horn. In turn, in animals treated with high doses of BPA, the number of such fibers was more than twice as high as those noted in the control animals and reached 53.89% in the right horn and 54.53% in the left horn. A clear BPA-induced increase in the number also concerned nNOS+/PACAP+ and nNOS+/SP+ nerves. Under the impact of low doses of BPA, the number of the former amounted to 37.07% of all nitrergic nerves in the right horn and 35.49% in the left horn. In turn, in animals receiving high doses of BPA, these values reached 47.41% in the right horn and 48.03% in the left horn. SP was found in 38.96% (right horn) and 36.78% (left horn) of all nNOS-LI nerves in animals receiving low doses of BPA, and 44.65% (right horn) and 42.54% (left horn) in pigs treated with high doses of this substance. The smallest changes concerned the degree of co-localization of nNOS with CART. The number of nNOS+/CART+ nerves increased to 22.33% (right horn) and 22.5% (in left horn) under the impact of low doses of BPA and to 29.43% (right horn) and 26.83% (left horn) under the impact of high doses of this substance. 

BPA also caused an increase in the number of nitrergic nerves located in the myometrium of the uterine horns, and the changes were more visible than those noted in the endometrium. In animals treated with low doses of BPA, the number of nNOS-LI nerves amounted to 4.38 nerves per observation field in the right horn and 4.82 in the left horn. Especially visible changes were noted under the impact of high doses of BPA. Namely, in this group of animals, the number of nitrergic nerves was 14.96 fibers per observation field in the right horn and 13.99 in the left horn. These values were more than ten times higher than those noted in the control animals.

BPA also caused an increase in the degree of co-localization of nNOS with all other substances studied in nerves located in the myometrium of the uterine horns (Table 2). Similar to the uterine corpus, the most visible changes concerned nNOS+/VIP+ and nNOS+/PACAP+ nerves. Under low doses of BPA, the number of nNOS+/VIP+ nerves increased to 39.6% of all nitrergic nerves in the right horn and 38.35% in the left horn. The administration of high doses of BPA resulted in more visible changes. Namely, in animals receiving high doses of BPA, the number of nNOS+/VIP+ nerves was about twice as high as that noted in control animals and amounted to 53.7% in the right horn and 54.3% in the left horn. In animals treated with low doses of BPA, PACAP was found in 39.02% of nitrergic nerves (right horn) and 39.85% (left horn), while in animals receiving high doses of BPA these values reached 52.04% (right horn) and 50.2% (left horn). 

BPA-induced changes concerning the co-localization of nNOS with SP were less visible, and the number of nNOS+/SP+ fibers reached 37.19% (right horn) and 39.34% (left horn) under the impact of low doses of BPA and 46.38% (right horn) and 45.82% (left horn) in animals receiving high doses of this substance. The fewest changes were noted for the nNOS+/CART+ nerves. In animals treated with low doses of BPA, CART was found in 28.94% of all nitrergic nerves in the right horn and 30.62% in the left horn. In turn, high doses of BPA caused an increase in the number of such nerves to 34.17% of all nNOS-LI fibers in the right horn and to 37.29% in the left horn. 

## 3. Discussion

During the present experiment, nitrergic nerve fibers were noted in the endometrium and myometrium of uterine corpus and horns both in physiological conditions and under the impact of both studied doses of BPA. The number of nNOS-LI fibers observed in the present investigation is in line with results obtained in the previous studies [10,11], which indicates that nitric oxide is an important neuronal factor involved in the regulatory processes within the uterus. In turn, the presence of nerves containing NO in various parts of the uterine wall confirms the multidirectional impact of this gaseous transmitter on the uterus, which is also known from previous publications, where the involvement of NO in uterine muscle activity, endometrial secretion, blood flow, and immunological processes has been reported [30,31]. The participation of nitrergic nerve fibers in various regulatory processes connected with the uterine activity may be also confirmed by a high degree of neurochemical diversification of such nerves and the co-localization of nNOS with a wide range of other neuronal substances, which was shown in the previous studies [10,11,21] and during the present investigation. It should be pointed out that substances noted in the present study in the uterine nitrergic nerves may be involved in various processes within the uterus, including the conduction of the sensory and pain stimuli, regulation of activity of the smooth muscles, as well as the influence on the endometrial secretion, blood flow in the uterine wall, immunological processes and the normal course of the estrus cycle [32]. 

Interestingly, some neuronal factors noted in the present study in the nNOS-LI nerves show the opposite activity. For example, VIP and PACAP in the light of previous studies (similarly to NO) are known as powerful factors exhibiting relaxant influence on the smooth muscles [33], whereas SP is described as a substance, which causes stimulation of the uterine contraction [34]. Thus, the issue of co-localization of nNOS and SP in the same nerves in the uterus and functions of such nerves in the regulation of smooth muscle activity is not clear. On the other hand, nerves simultaneously containing NO and SP noted in the present study may be sensory structures, because the participation of both of these substances in sensory stimuli conduction has been previously reported [35].

The important aspect connected with the functions of NO within the uterine reported by previous investigations is the participation in adaptive processes, being a response to both physiological changes (connected for example with changes occurring during pregnancy) and pathological processes [36,37]. Such functions of uterine nitrergic nerves are confirmed by the fact that all substances co-localized with nNOS noted in the present investigation have been described previously as factors showing important roles in protective and adaptive reactions during pathological states concerning both the reproductive system and autonomic nervous structures supplying other internal organs [38]. 

Such adaptive and protective functions of nitrergic nerves located in the uterine wall are also confirmed by the BPA-induced changes in the number and neurochemical characterization of nNOS-LI fibers noted in the present investigation. Interestingly, these changes have been observed even under the impact low doses of BPA—at the level of 0.05 mg/kg b.w./day, and it should be underlined that this dose is established by legal provisions of some countries as a tolerable daily intake (TDI) or reference dose for BPA and it is, therefore, considered to be safe for humans and animals [39]. However, the present observations clearly indicate that even relatively short (lasting four weeks) exposure to such low doses of BPA is not completely neutral to an organism, because it is capable of changes in the expression of neuronal substances in nerves located in the uterine wall. These results, together with previous studies, where the influence of low doses of BPA on the autonomic nervous system supplying various internal organs has been reported [40], strongly suggest that revision of BPA doses that are completely safe for humans and animals is needed. Incidentally, the European Food Safety Authority, the agency of the European Union engaged in risks associated with the food chain, decreased TDI of BPA from 0.05 mg to 4 µg/kg b.w./day [41], and the results of toxicological studies justify this decision [42]. It should also be noted that changes in neurochemical characterization of nervous structures under the impact of BPA may be the first signs of intoxication with this substance, which is suggested not only by these studies but also by previous investigations [39,40]. 

The changes in the number and neurochemical characterization of nitrergic nerves found during the present study may result from the neurotoxic impact of BPA, as well as by its pro-inflammatory properties. 

The neurotoxic influence of BPA is relatively well known. Exposure to this substance results in inhibition and disturbances in the development of neuronal cells, especially dendrites, axons, and synapses [43]. Moreover, BPA causes abnormalities in the transcellular ion transport and synthesis of a wide range of neuronal active substances playing the roles of neuromodulators and/or neuromodulators [44], which, in extreme cases, may lead to dysfunctions in learning and memory as well as neurodegenerative diseases [45]. 

However, it is known that neuronal tissue in response to pathological and toxic factors demonstrates “neuroplasticity”, which is understood as the capacity to undertake morphological and physiological changes in order to ensure neuroprotection and homeostasis maintenance [46]. Changes noted in the present study may be manifestations of such neuroprotective and adaptive processes. Although in the case of NO, the matter of participation in neuroprotective reactions seems to be debatable. Although it is known that this gaseous neurotransmitter may be involved in some protective processes in various parts of the nervous system [47,48], it should not be forgotten that this substance produced in excess by cells under the impact of pathological factors is a free radical, which shows cytotoxic properties [49]. 

However, the thesis that the observed changes are connected with protective and adaptive processes is supported by the fact that all other neuronal substances investigated in the present study are known as factors involved in such reactions. Neuroprotective properties are best known in the case of VIP, which affects the Schwann cells and promotes neuronal survivability under the impact of pathological, toxic, and mechanical factors [50]. In the case of other substances, the knowledge concerning this issue is more fragmentary. However, despite this, it is known that CART protects neurons against focal cerebral ischemia and neurodegeneration occurring during Parkinson’s disease [51], SP defends neurons against apoptosis and death caused by ionic disorders [52], and PACAP plays a protective role in reactions connected with oxidative stress [53]. 

Although the majority of data on the neuroprotective properties of CART, SP, and PACAP are limited to the central nervous system, it cannot be excluded that these substances show similar activity in the peripheral nervous system, which is also suggested in the previous studies [54]. Moreover, the fact that the most visible changes noted in the present investigation concern VIP (the most potent neuroprotective factor), this strongly suggests that the neuroprotective processes are the basis in observed fluctuations in the neurochemical characterization of nerves in the uterus.

Another cause of changes in the number and neurochemical coding of nerve fibers identified in the present study may be the pro-inflammatory properties of BPA, which influence the number of T and B lymphocytes and levels of pro-inflammatory cytokines, including cytokines IL-1β, IL-6, and TNF-α [55]. However, all neuronal substances included in the present study are involved in pro- or anti-inflammatory reactions. Strong anti-inflammatory activity is characteristic for VIP and PACAP, which inhibit the synthesis of pro-inflammatory factors [56], while SP is known as a substance showing pro-inflammatory activity affecting the lymphocytes and macrophages and causing an increase in the level of TNF-α [57]. In turn, the participation of NO in inflammatory processes is not quite clear. According to previous studies, NO may show both pro- and anti-inflammatory activity, which depends on the type of tissue affected by inflammation and the direct reason of the inflammatory process [58]. Interestingly, during the present study, BPA caused an increase in the number of fibers immunoreactive to both anti- and pro-inflammatory neuronal factors. The mechanisms underpinning this phenomenon are not known, but a similar situation was observed during studies concerning the influence of BPA on the innervation supplying the other internal organs [59].

It should be underlined that the changes noted in the present study, especially those observed within the nerves located in the myometrium, may result from the direct impact of BPA on the smooth muscles. It is known that this substance is a potent relaxant factor affecting, among others, muscles located in the gastrointestinal tract and uterus and causing a decrease in the amplitude and frequency of smooth muscle contraction [60]. Moreover, the relaxant activity of BPA within the uterus is done by a nitrergic mechanism [61], which would explain the increase in the number of nitrergic nerves noted in the present study. Moreover, other neuronal substances included in the study are also involved in regulatory processes connected with the contraction of the smooth muscles [62], which also suggests that the observed changes are connected with the influence of BPA on muscular contractility.

## 4. Materials and Methods

Fifteen immature female pigs of the Piétrain x Duroc race at the age of 8 weeks were used during the present study. During the experiment, animals were kept in pens (5 animals per pen) appropriate for age and animal species. They were fed the commercial feed for piglets and had unrestricted access to drinking water. All procedures were performed in accordance with the guidelines of the Local Ethical Committee responsible for experimental animals in Olsztyn (Poland) (decision numbers 28/2013 of 22 May, 2013 and 65/2013/DLZ of 27 November, 2013).

At the beginning of the experiment, the animals were subjected to a five-day adaptation period and divided into three groups (five animals in each): control (C) group, experimental I (Ex I) group, and experimental II (Ex II) group. Animals of all groups received gelatin capsules orally once daily before the morning feeding in the following manner: control animals received empty capsules, animals of Ex I group were treated with capsules with BPA (BISPHENOL A, catalog no. 239658, Sigma Aldrich, Poznan, Poland) in a dose of 0.05 mg/kg body weight (b.w.)/day, and pigs of Ex II group received capsules with BPA in a dose ten times higher (0.5 mg/kg b.w./day). The lower dose of BPA used in the present study is a dose that, in many countries, is recognized as a tolerable daily intake (TDI) or reference dose for BPA and is, therefore, considered to be safe for humans and animals [63]. In turn, the higher dose of BPA used in the present study is a dose where changes in the neurochemical characterization are visible in the nervous structures supplying the gastrointestinal tract [39].

After 28 days of administration of BPA, all animals were pre-medicated with Stresnil (Janssen, Belgium, 75 μL/kg of b.w., IM) and euthanized using an overdose of sodium thiopental (Thiopental, Sandoz, Kundl, Austria) given intravenously to the marginal ear vein. 

After euthanasia, the uteri were collected from all animals. Immediately after collection, organs were fixed in 4% buffered paraformaldehyde (pH 7.4) for one hour and rinsed in a phosphate buffer for three days with a daily exchange of buffer. After three days, the uteri were put into 18% phosphate-buffered sucrose and stored at 4 °C for at least three weeks. Fragments of the uterine corpus (c. 0.5 cm long) located just below the uterine bifurcation and fragments of left and right uterine horns (c. 0.5 cm long) positioned about 2 cm above the place where the uterine corpus transitions into horns) were then frozen at −22 °C and cut into 14 μm-thick sections with the cryostat (HM 525, Microm International, Germany). Slices of the uteri fragments were mounted on microscopic slides and stored at −20 °C until further studies. 

Fragments of the uteri were subjected to standard single and double immunofluorescence technique according to the method described previously by Rytel [21], in which commercial antibodies against neuronal isoform of nitric oxide synthase (nNOS—used here as a marker of nitrergic neurons), vasoactive intestinal polypeptide (VIP), substance P (SP), pituitary adenylate cyclase-activating peptide (PACAP), and cocaine- and amphetamine-regulated transcript (CART) were used (Table 1). In short, the labeling procedure consisted of the following stages (all stages were performed at room temperature, stages 2, 3, and 4 were performed in the humid chamber): (1) drying the slices for 1 h; (2) incubation with blocking solution (10% normal goat serum, 0.1% bovine serum albumin, 0.01% NaN_3_, 0.25% Triton x-100 and 0.05% thimerosal in PBS) for 1 h; (3) incubation with one (single immunofluorescence technique) or a mixture of two (double immunofluorescence technique) primary antibodies for 20 h; (4) incubation with species-specific secondary antibodies marked with various fluorochromes (Table 1); (5) treatment with buffered glycerol and coverage with coverslips. Between each stage mentioned above, the labeling slide with uterine fragments was rinsed for ten minutes with three changes of 0.01 M PBS pH 7.2. 

To confirm the specificity of antibodies, routine tests were used, including pre-absorption of the antisera with appropriate antigen and omission, and replacement tests were performed.

After labeling, fragments of the uteri were viewed under an Olympus BX51 microscope under appropriate immunofluorescence filter sets (Olympus, Tokyo, Japan). During the viewing, the density of nNOS-like immunoreactive (nNOS-LI) nerves located in the mucosal and muscular layer of the uterine corpus and both horns were evaluated. This evaluation consisted of counting such nerve fibers *per* microscopic observation field (0.1 mm^2^). The counting was performed in five randomly selected microscopic observation fields in the endometrium and myometrium of six sections of the uteri (30 observation fields of endometrium and myometrium of each part of the uterus from each animal used in the investigation). To prevent double-counting of the same nerve structures, slices of the uteri included in the evaluation were located at least 200 µm apart.

The evaluation of the neurochemical characterization of nerve fibers immunoreactive to nNOS was conducted differently. To determine the degree of co-localization of nNOS with other neuronal active substances in the same nerves, at least 500 nitrergic nerves located in the endometrium and myometrium of corpus and both horns of the uterus from each animal under investigation were examined for the presence of each neuronal factor studied. The degree of co-localization of nNOS with other substances was presented as the percentage of nerves immunoreactive simultaneously to nNOS and VIP, nNOS and SP, nNOS and PACAP and nNNOS and CART with regard to the total number of nerves containing nNOS, and nNNOS-LI nerve fibers, which were considered as representing 100%. The slices of the uteri included in the evaluation of neurochemical characterization of nitrergic neurons were also located at least 200 µm apart to avoid double-counting the same nerves. 

The statistical analysis was performed with a one-way analysis of variance (ANOVA) with Bonferroni’s Multiple Comparison post hoc test using Statistica 9 software (Statistica 9.1, StatSoft, Inc., Cracow, Poland). Differences were considered significant at *p* < 0.05 (*).

## 5. Conclusions

The results obtained in the present study clearly indicate that nitrergic nerve fibers play important roles in the regulation of the uterine functions and show considerable differentiation in terms of neurochemical characterization. Moreover, the present investigation has shown that even low doses of BPA result in changes both in the number of nitrergic nerves in the uterine wall, as well as the degree of co-localization of NO with other neuronal substances. The observed changes are probably a response to the neurotoxic and/or pro-inflammatory activity of BPA and are connected with protective processes. Nevertheless, the exact mechanisms underlying the observed changes are not completely clear.

## Figures and Tables

**Figure 1 ijms-21-04543-f001:**
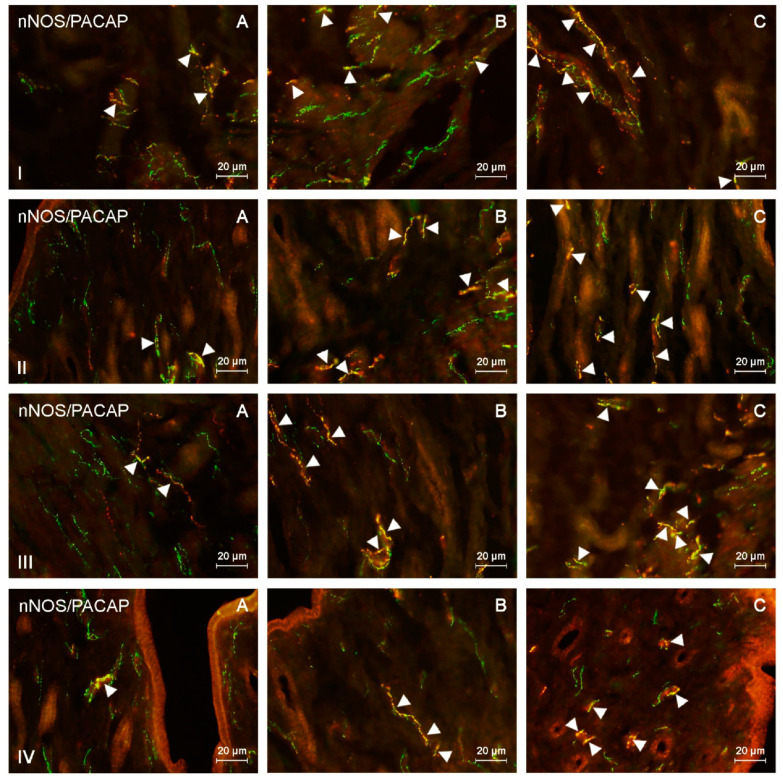
Neuronal nitric oxide synthase (nNOS)-positive nerves (nNOS+ green) immunoreactive to pituitary adenylate cyclase-activating peptide (PACAP) (red) in the: (I) muscular layer of the corpus, (II) mucosal layer of the corpus, (III) muscular layer of the horns, (IV) mucosal layer of the horns of the uterine body of control animals (**A**) and pigs treated with low (**B**) and high (**C**) dose of bisphenol A. Nerves simultaneously immunoreactive to nNOS and PACAP (yellow) are indicated with arrowheads.

**Figure 2 ijms-21-04543-f002:**
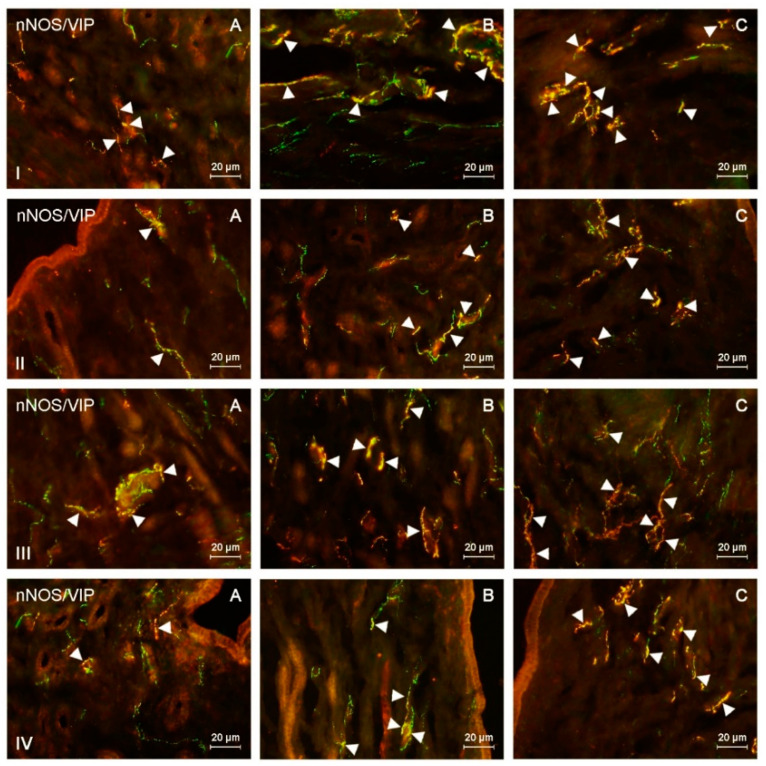
nNOS-positive nerves (nNOS+ green) immunoreactive to vasoactive intestinal polypeptide (VIP) (red) in the: (I) muscular layer of the corpus, (II) mucosal layer of the corpus, (III) muscular layer of the horns, (IV) mucosal layer of the horns of the uterine body of control animals (**A**) and pigs treated with low (**B**) and high (**C**) dose of bisphenol A. Nerves simultaneously immunoreactive to nNOS and VIP (yellow) are indicated with arrowheads.

**Figure 3 ijms-21-04543-f003:**
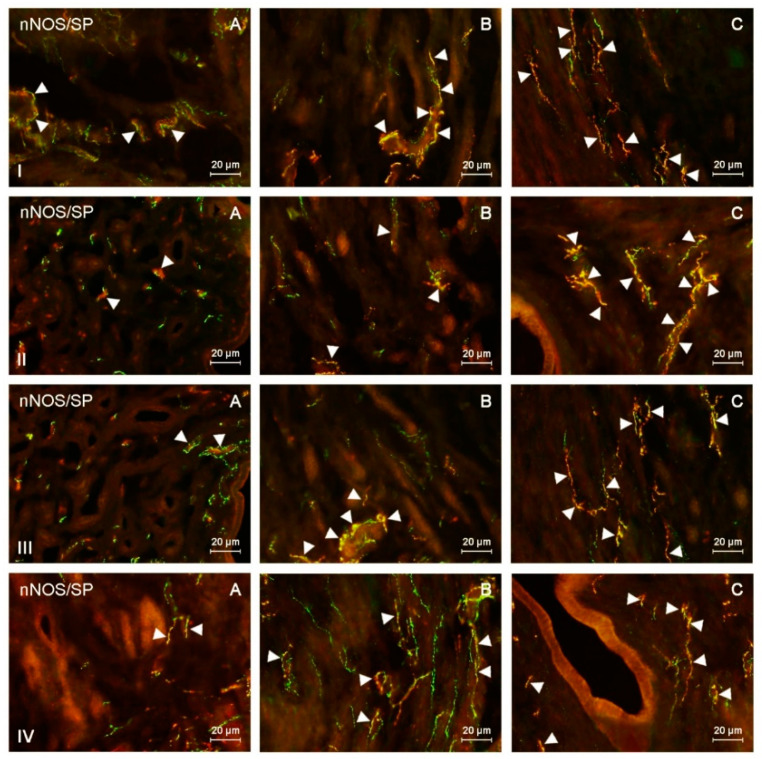
nNOS-positive nerves (nNOS+ green) immunoreactive to substance P (SP) (red) in the: (I) muscular layer of the corpus, (II) mucosal layer of the corpus, (III) muscular layer of the horns, (IV) mucosal layer of the horns of the uterine body of control animals (**A**) and pigs treated with low (**B**) and high (**C**) dose of bisphenol A. Nerves simultaneously immunoreactive to nNOS and SP (yellow) are indicated with arrowheads.

**Figure 4 ijms-21-04543-f004:**
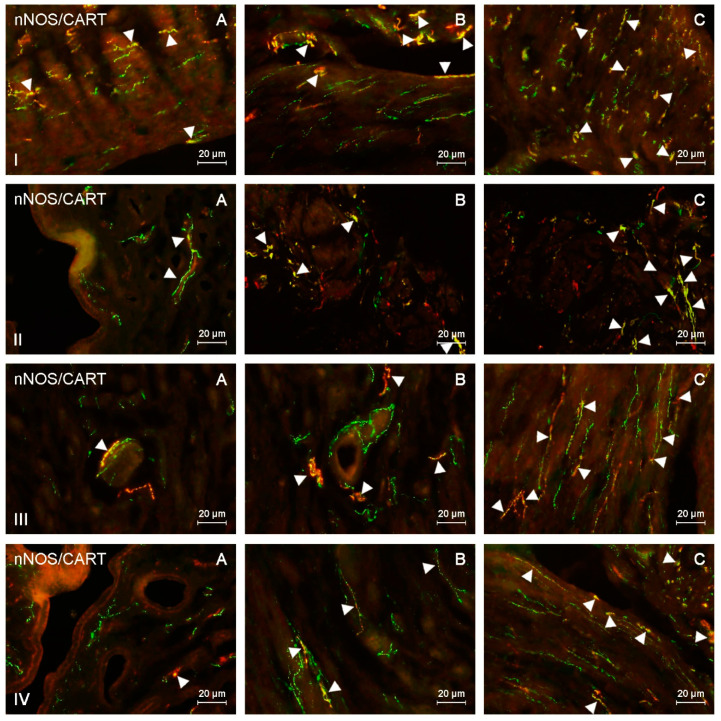
nNOS-positive nerves (nNOS+ green) immunoreactive to cocaine- and amphetamine-regulated transcript (CART) (red) in the: (I) muscular layer of the corpus, (II) mucosal layer of the corpus, (III) muscular layer of the horns, (IV) mucosal layer of the horns of the uterine body of control animals (**A**) and pigs treated with low (**B**) and high (**C**) dose of bisphenol A. Nerves simultaneously immunoreactive to nNOS and CART (yellow) are indicated with arrowheads.

**Table 1 ijms-21-04543-t001:** The number of nNOS-positive fibers and their neurochemical characterization in the uterine corpus as well as in the rights and lefts horns in muscular layer of the control animals (CTRL) and pigs treated with low (E1) and high (E2) dose of bisphenol A.

Muscular Layer									
	Corpus of the Uterus			Right Horn			Left Horn		
	CTRL	ExI	ExII	CTRL	ExI	ExII	CTRL	ExI	ExII
nNOS+	4.74 ± 0.62	5.52 ± 0.67 ^b^	13.5 ± 1.06 ^b^	1.54 ± 0.30 ^a,b^	4.38 ± 0.47 ^a,c^	14.96 ± 0.73 ^b,c^	1.66 ± 0.30 ^a,b^	4.82 ± 0.65 ^a,c^	13.99 ± 0.74 ^b,c^
nNOS+/PACAP	30.83 ± 1.85 ^a,b^	48.65 ± 1.75 ^a,c^	57.78 ± 2.83 ^b,c^	28.5 ± 1.52 ^a,b^	39.02 ± 2.09 ^a,b^	52.04 ± 2.30 ^b,c^	27.22 ± 1.33 ^a,b^	39.85 ± 1.86 ^a,b^	50.2 ± 2.92 ^b,c^
nNOS+/VIP+	34.91 ± 1.09 ^a,b^	46.22 ± 1.24 ^a,c^	62.53 ± 2.60 ^b,c^	25.97 ± 1.33 ^a,b^	39.6 ± 1.97 ^b,c^	53.7 ± 1.86 ^b,c^	25.99 ± 2.20 ^a,b^	38.35 ± 1.43 ^b,c^	54.3 ± 1.23 ^a,b^
nNOS+/SP+	40.33 ± 1.73 ^b^	44.76 ± 1.48 ^c^	54.74 ± 2.47 ^b,c^	30.44 ± 1.40 ^b^	37.19 ± 1.58 ^c^	46.38 ± 3.33 ^b,c^	30.53 ± 1.12 ^a,b^	39.34 ± 2.91 ^a^	45.82 ± 2.04 ^b^
nNOS+/CART+	24.67 ± 1.66 ^a,b^	32.46 ± 1.94 ^a^	36.25 ± 1.24 ^b^	26.11 ± 1.36 ^b^	28.94 ± 2.93	34.17 ± 3.12 ^b^	25.43 ± 2.61 ^b^	30.62 ± 2.77	37.29 ± 2.55 ^b^

(1) The average number of fibers in the microscopic observation field (0.1 mm^2^). (2) The percentage of nerves immunoreactive to the particular substances in respect to all NOS-positive nerves (NOS-positive nerves were considered as representing 100%). Statistically significant data (*p* ≤ 0.05) in particular rows are marked by different letters, and not significant data are marked by the same letters.

**Table 2 ijms-21-04543-t002:** The number of nNOS-positive fibers and their neurochemical characterization in the uterine corpus as well as in the rights and lefts horns in mucosal layer of the control animals (CTRL) and pigs treated with low (ExI) and high (ExII) dose of bisphenol A.

Mucosal Layer									
	Corpus of the Uterus			Right Horn			Left Horn		
	CTRL	ExI	ExII	CTRL	ExI	ExII	CTRL	ExI	ExII
nNOS+	5.19 ± 2.20 ^b^	4.71 ± 0.58 ^c^	13.07 ± 1.06 ^b,c^	1.54 ± 0.41 ^a,b^	3.43 ± 0.41 ^a,c^	4.17 ± 0.69 ^b,c^	1.65 ± 0.60 ^a,b^	3.63 ± 0.50 ^a,c^	4.38 ± 0.59 ^b,c^
nNOS+/PACAP	32.56 ± 1.91 ^a,b^	42.11 ± 2.17 ^a,c^	60.27 ± 2.21 ^b,c^	26.42 ± 1.64 ^a,b^	37.07 ± 1.45 ^a^	47.41 ± 2.99 ^b^	25.52 ± 2.12 ^a,b^	35.49 ± 1.6 ^a^	48.03 ± 2.26 ^b^
nNOS+/VIP+	29.07 ± 1.52 ^a,b^	44.46 ± 1.87 ^a,c^	64.13 ± 1.65 ^b,c^	24.33 ± 2.4 ^a,b^	35.28 ± 1.62 ^b,c^	53.89 ± 2.40 ^b,c^	25.21 ± 1.97 ^a,b^	36.51 ± 1.47 ^b,c^	54.53 ± 1.27 ^b,c^
nNOS+/SP+	31.94 ± 2.34 ^a,c^	39.1 ± 2.36 ^b,c^	50.64 ± 3.40 ^b,c^	31.05 ± 1.89 ^a,b^	38.96 ± 1.96 ^a^	44.65 ± 2.66 ^b^	31.56 ± 3.29 ^a,b^	36.78 ± 2.62 ^a^	42.54 ± 3.00 ^b^
nNOS+/CART+	29.2 ± 1.45 ^b^	31.92 ± 1.78	36.08 ± 1.61 ^b^	19.04 ± 0.91 ^b^	22.39 ± 0.91	29.43 ± 1.11 ^b^	18.74 ± 1.21 ^b^	22.5 ± 0.99	26.93 ± 1.06 ^b^

(1) The average number of fibers in the microscopic observation field (0.1 mm^2^). (2) The percentage of nerves immunoreactive to the particular substances in respect to all NOS-positive nerves (NOS-positive nerves were considered as representing 100%). Statistically significant data (*p* ≤ 0.05) in particular rows are marked by different letters, and not significant data are marked by the same letters.

**Table 3 ijms-21-04543-t003:** List of antisera and reagents used in immunohistochemical investigations.

**Primary Antibodies**				
**Antigen**	**Code**	**Species**	**Working Dilution**	**Supplier**
nNOS	N218	Rabbit	1:1000	Sigma Aldrich, Darmstadt, Germany
SP	8450-0505	Rat	1:1000	BioRad Hercules, CA, USA
VIP	9535-0504	Mouse	1:2000	Biogenesis Ltd., Poole, GB
PACAP	T-5039	Guinea pig	1:8000	Peninsula Labs, San Carlos, CA, USA
CART	MAB 163	Mouse	1:6000	R&D Systems, Minneapolis, USA
**Secondary Antibodies**				
**Reagents**		**Code**	**Working Dilution**	**Supplier**
Alexa fluor 488 donkey anti-rabbit IgG		A21206	1:1000	ThermoFisher Scientific, Waltham, MA, USA
Alexa fluor 546 donkey anti-mouse IgG		A10036	1:1000	ThermoFisher Scientific, Waltham, MA, USA
Alexa fluor 546 donkey anti-guinea pig IgG		A-11074	1:1000	ThermoFisher Scientific, Waltham, MA, USA
Alexa fluor 546 donkey anti-rat IgG		A11081	1:1000	ThermoFisher Scientific, Waltham, MA, USA

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
