# Peer review of "The Influence of Bisphenol a on the Nitrergic Nervous Structures in the Domestic Porcine Uterus"

_ijms, 2020, doi:10.3390/ijms21124543_

Round 1

Reviewer 1 Report

The article "The influence od bisphenol a on the nitrergic nervous structures in the Uterus" is quite well conceived, however several modifications should be made to improve it:

-The order of subsection is not campatibel with the requirment of IJMS

-why the authors used these and not other doses of BPA ? it should be clarified in "Material and Methods"

-Table 1must be suplemmented with additional information, including town and country of producer, dilutions of antibodies and  animal species of secondary antiodies

-In "Materials and methods" the authors wrote that animals were divided into control (C), experimental I and II groups (Ex I, Ex II), but in tables 1 and 2 results have been obtained from E1 and E2, plaese correct it

-The authors wrote that microscope was equipped with "appopriate IF filter sets". The exact type of filters should be mentioned.

-Tables 2 and 3 are not readable. The letters and digits are to small. Tables should be reedited.

-A scale bar should be added into each photograph

-A complete, extensively Revision of the English lanuage should be performed

Author Response

The authors thank reviewers for insightful reviews, which have contributed to improving the quality of the manuscript. All suggestions of the Reviewers were taken into account.

-why the authors used these and not other doses of BPA ? it should be clarified in "Material and Methods"

The text has been supplemented by: „The lower dose of BPA used in the present study is a dose which, in many countries, is recognized as a tolerable daily intake (TDI) or reference dose for BPA and is therefore considered to be safe for humans and animals [63]. In turn, the higher dose of BPA used in the present study is a dose where changes in the neurochemical characterization are clearly visible in the nervous structures supplying the gastrointestinal tract [39]”.

- Table 1must be suplemmented with additional information, including town and country of producer, dilutions of antibodies and  animal species of secondary antiodies

-Tables 2 and 3 are not readable. The letters and digits are to small. Tables should be reedited.

The table has been reedited.

-In "Materials and methods" the authors wrote that animals were divided into control (C), experimental I and II groups (Ex I, Ex II), but in tables 1 and 2 results have been obtained from E1 and E2, plaese correct it

The materials and methods has been corrected.

-The authors wrote that microscope was equipped with "appopriate IF filter sets". The exact type of filters should be mentioned.

-A scale bar should be added into each photograph

The authors, according to the Reviewer’s suggestion, added scale bar and more information about filters.

-A complete, extensively Revision of the English lanuage should be performer

The text of the manuscript has been edited  as regards language style by native speaker – worker of specialized translational office, specialist in the field of biological sciences.

The authors hope that corrections of the text and explanations will allow to publish the manuscript in “International Journal of Molecular Science”.

Reviewer 2 Report

Title: The influence of bisphenol a on the nitrergic nervous  structures in the uterus

Authors : Rytel L and Gonkowski S.

Overview:

The aim of this study was to examine  the  influence of various doses of BPA of nitrergic nerves supplying the uterus with double immunofluorescence method.

The present study documented the specific NO, SP, VIP, PACAP and CART expression patterns  under the influence of BPA (under the impact low and a high dose of bisphenol)

This study shows, for the first time,  that even low doses of BPA result in the changes both in the number of nitrergic nerves in the uterine wall, as well  as the degree of co-localization of NO with other neuronal substances.

Authors suggest that  observed changes  are the answer to the neurotoxic and/or pro-inflammatory activity of BPA and are connected with  protective processes.

In general, the methods used are appropriate, and appear to have been carefully carried out.

Specific comments:

  1. The title of the manuscript must be reedited. Authors studied distribution of nerves in the domestic pig and this fact must be highlighted in the title
  2. Lists of abbreviation used in the text added at the beginning of the manuscript would increase its readability
  3. Figure legends are not clear. They mentioned photographs a, b and c, while on photographs letters A, B and C are presented. Figure legenda are too long. I suggest to reduce them (if it is possible)
  4. Some values in tables 2 and 3 and in the text are given to two decimal places, and other only to one decimal place. Why?
  5. Size units of values presented in tables 2 and 3 must be added in legends.
  6. Relatively large number of references cited in the text are out of dates (they have been published 15-20 years ago). They should be replaced by new research results in recent year.
  7. There are many spelling and grammatical mistakes in the text. It is better to be checked by English native speaker

Author Response

The authors thank reviewers for insightful reviews, which have contributed to improving the quality of the manuscript. All suggestions of the Reviewers were taken into account.

   -The title of the manuscript must be reedited. Authors studied distribution of nerves in the domestic pig and this fact must be highlighted in the title

The title has been changed accoring to the Reviewers’s suggestion.

    Lists of abbreviation used in the text added at the beginning of the manuscript would increase its readability

The list of abberevation has been added

- Figure legends are not clear. They mentioned photographs a, b and c, while on photographs letters A, B and C are presented. Figure legenda are too long. I suggest to reduce them (if it is possible)

- Some values in tables 2 and 3 and in the text are given to two decimal places, and other only to one decimal place. Why?

- Size units of values presented in tables 2 and 3 must be added in legends.

This part of the manuscript has been thoroughly reedited

    Relatively large number of references cited in the text are out of dates (they have been published 15-20 years ago). They should be replaced by new research results in recent year.

The oldest citation has been repleced by researarch pulished in last years

    There are many spelling and grammatical mistakes in the text. It is better to be checked by English native speaker

The text of the manuscript has been edited  as regards language style by native speaker – worker of specialized translational office, specialist in the field of biological sciences.

The authors hope that corrections of the text and explanations will allow to publish the manuscript in “International Journal of Molecular Science”.